# *Trichoderma longibrachiatum* and *Trichoderma asperellum* Confer Growth Promotion and Protection against Late Wilt Disease in the Field

**DOI:** 10.3390/jof7060444

**Published:** 2021-06-02

**Authors:** Ofir Degani, Onn Rabinovitz, Paz Becher, Asaf Gordani, Assaf Chen

**Affiliations:** 1Migal Galilee Research Institute, Tarshish 2, Kiryat Shmona 11016, Israel; onnrab@gmail.com (O.R.); pazbec@gmail.com (P.B.); asigordani1@gmail.com (A.G.); assafc@migal.org.il (A.C.); 2Faculty of Sciences, Tel-Hai College, Upper Galilee, Tel-Hai 12210, Israel

**Keywords:** biological control, *Cephalosporium maydis*, crop protection, field assay, fungus, *Harpophora maydis*, *Magnaporthiopsis maydis*, maize, real-time PCR

## Abstract

Late wilt disease (LWD) of maize, caused by *Magnaporthiopsis maydis*, is considered a major threat to commercial fields in Israel, Egypt, Spain, and India. Today’s control methods include chemical and agronomical intervention but rely almost solely on resistant maize cultivars. In recent years, LWD research focused on eco-friendly biological approaches to restrain the pathogen. The current study conducted during two growing seasons explores the potential of three *Trichoderma* species as bioprotective treatments against LWD. These species excelled in preliminary assays performed previously under controlled conditions and were applied here in the field by directly adding them to each seed with the sowing. In the first field experiment, *Trichoderma longibrachiatum* successfully rescued the plants’ growth indices (weight and height) compared to *T.* *asperelloides* and the non-treated control. However, it had no positive effect on yield and disease progression. In the subsequent season, this *Trichoderma* species was tested against *T. asperellum,* an endophyte isolated from susceptible maize cultivar. This experiment was conducted during a rainy autumn season, which probably led to a weak disease burst. Under these conditions, the plants in all treatment groups were vivid and had similar growth progression and yields. Nevertheless, a close symptoms inspection revealed that the *T. longibrachiatum* treatment resulted in a two-fold reduction in the lower stem symptoms and a 1.4-fold reduction in the cob symptoms at the end of the seasons. *T.* *asperellum* achieved 1.6- and 1.3-fold improvement in these parameters, respectively. Quantitative Real-time PCR tracking of the pathogen in the host plants’ first internode supported the symptoms’ evaluation, with 3.1- and 4.9-fold lower *M. maydis* DNA levels in the two *Trichoderma* treatments. In order to induce LWD under the autumn’s less favorable conditions, some of the plots in each treatment were inoculated additionally, 20 days after sowing, by stabbing the lower stem section near the ground with a wooden toothpick dipped in *M. maydis* mycelia. This infection method overrides the *Trichoderma* roots protection and almost abolishes the biocontrol treatments’ protective achievements. This study suggests a biological *Trichoderma*-based protective layer that may have significant value in mild cases of LWD.

## 1. Introduction

Late-wilt disease (LWD) is a severe corn disease that results in the drying of sweet and fodder corn varieties before harvest. The cause of the disease, the soil- and seed-borne fungus *Magnaporthiopsis maydis* [1], formerly known as *Harpophora maydis* and *Cephalosporium maydis,* was discovered in Egypt [2] and is now reported in eight countries and causes substantial economic losses in Egypt [3], India [4], Spain, Portugal [5] and Israel [6]. The pathogen’s survival depends on soils with infected host remains [7], infected seeds [8], or alternative hosts such as *Lupinus termis* (lupine) [9], cotton, watermelon, and *Setaria viridis* (green foxtail) [10,11].

Today, the primary means of protecting crops against LWD is the use of resistant maize varieties [12]. At the same time, the immunity of these varieties requires validation to be covered by insurance, and they are often of lower market value than the susceptible varieties. Furthermore, resistant maize varieties are under constant threat of developing violent pathogenic variations of the fungus, as reported in Egypt and Spain [13,14], and may lose their immunity, which occurred recently in Israel in the sweet Royalty and fodder 32D99 corn cultivars. Moreover, the pathogen is hiddenly (without symptoms) established in resistant plants, and their seeds allow the disease to spread [6].

A recent focused research effort yielded progress in our ability to restrain LWD chemically. We now have an effective and economic azoxystrobin-based prevention protocol [15,16,17,18]. Still, the method’s application requires dripline irrigation for coupled rows, and there is a constant risk of resistance developing [15,19]. In addition, chemical preparations may have a widespread environmental influence (such as on beneficial microorganisms in the soil), and they are considered a health risk. Biological pesticides are environmentally friendly; hence this approach occupies an increasingly central place in global scientific research towards this end.

Although this scientific direction has been widely explored over the years against many destructive phytopathogens [20], many knowledge gaps regarding LWD still exist. Thus, the potential of eco-friendly solutions for LWD has only now started to be revealed. Biological control using *Trichoderma* spp. or other microorganisms against the LWD causal agent has been demonstrated in several studies (most recently [21,22]). Members in this genus can form endophytic mutualistic associations with plant species [23], while others have developed as bioprotective agents against fungal phytopathogens [24]. It was later shown [22] that microalgae, *Chlorella vulgaris* extracts with *T. virens* and *T. koningii*, were effective treatments against LWD under greenhouse and field conditions. Another approach uses plant growth-promoting rhizobacteria as a biological control against LWD associated with the maize rhizosphere, which could also improve plant health (summarized by [21]).

We recently scanned three *Trichoderma* isolates with inhibitory activity against *M. maydis*: *T. asperelloides* (T.203); *T. longibrachiatum* (T.7407 from marine source [25]); and *T. asperellum* (P1), an endophyte isolated in our laboratory from corn seeds of a strain susceptible to LWD [26]. These isolates prevented the pathogen’s growth in culture plates, significantly reducing its establishment and development in seedlings’ corn plant tissues and resulting in significant improvement in growth and crop indices in potted plants under field conditions [27]. To establish these isolates’ bioprotective potential for commercial corn production, the current study focuses on implementing them in LWD-infested maize fields in a two-season study. Recently practiced quantitative real-time PCR (qPCR) was applied to the study of the pathogenesis and diagnosis of the effect of the *Trichoderma* preventive treatment.

## 2. Materials and Methods

### 2.1. Origin and Growth of Magnaporthiopsis maydis

All *M. maydis* isolates selected for this study (Table 1) were from our isolates library. The isolates were previously recovered from cornfields in the Hula Valley in Upper Galilee, northern Israel, from LWD-susceptible plants showing dehydration symptoms. The Israeli *M. maydis* isolates (including all strains selected for this research) were previously characterized and identified by their pathogenicity, physiology, colony morphology, and microscopic traits [6,28,29]. The microscopic and morphological characteristics of the pathogen were similar to strains previously described in Egypt and India [4,30]. Final confirmation of these strains was achieved using PCR-based DNA analysis [6,31]. The fungal colonies were grown on rich potato dextrose agar (PDA; Difco Laboratories Detroit, Detroit, MI, USA) at a temperature of 28 ± 1 °C in the dark under high humidity. Transferring the fungus to a new plate was carried out by moving a 6-mm (in diameter) colony agar disk to a new PDA containing a Petri dish. Fungus agar disks were cut from the margins of a culture of *M. maydis*, which was grown on PDA for 4–6 days. Petri dishes were maintained in a 28 °C incubator in the dark. Growth of submerged cultures was carried out using 10 fungal discs sown in an Erlenmeyer flask containing 150 mL potato dextrose broth (PDB, Difco Laboratories Detroit, Detroit, MI, USA). The flasks were sealed with a breathable plug and incubated for six days (with shaking at 150 rpm) at 28 ± 1 °C in the dark.

### 2.2. Origin and Growth of the Trichoderma Species

The three *Trichoderma* spp. inspected in this study were obtained from different sources (Table 1). *T. longibrachiatum* (isolate T7407) isolated from Mediterranean sponge *Psammocinia* sp. and the well-established biocontrol strain T203 (*T. asperelloides* [32]) were received courtesy of Prof. Oded Yarden (Hebrew University of Jerusalem, Israel) and were previously characterized [25]. The *Trichoderma asperellum* (isolate P1) is a fungal endophyte isolated from Prelude cv. (sweet maize from SRS Snowy River seeds, Australia, supplied by Green 2000 Ltd., Bitan Aharon, Israel) grains, and identified as previously described ([26]). All *Trichoderma* isolates were previously tested in a series of experiments in the lab, and sprouts were selected for the current study based on their high performance in these tests (their strong bioprotective activity against *M. maydis*) [26,27]. The growth conditions of the *Trichoderma* isolates were similar to the *M. maydis* growth conditions described above.

### 2.3. Overall Description of the Field Experiments

This study examined the biocontrol potential of selected *Trichoderma* isolates against *M. maydis* over a whole growth period in two subsequent experiments aimed at commercial field conditions simulation. Both experiments were performed on the Gadash experimental farm, Hula Valley, Upper Galilee, northern Israel (33°10′48.0′′ N 35°35′11.6′′ E) during the summer and fall of 2019 and 2020. The meteorological data recorded during the growing seasons are detailed in Table 2. As expected, the average temperature and soil temperature, radiation, and evaporation measurements in the 2019 summer experiment were higher. The average meteorological parameters measured during the 2020 experimental period were not optimal for the LWD burst because of the late season, which led to a lower temperature, radiation and evaporation, and higher precipitation (detailed in Table 2).

The two subsequent experiments had a similar experimental design. The experiment field has a record of moderate LWD infection. We deliberately inoculated the experiment plots (except the control plots) to achieve a more severe disease burst. Thus, the negative control in the experiments was plots without enrichment with the pathogen. All treatments were sown with Prelude cv. seeds, which were previously proven to be highly susceptible to late wilt disease [17]. Seeds were pre-treated with thiram, captan, carboxin, metalaxyl-M (manufactured by Rogers/Syngenta Seeds, Boise, ID, USA, supplied by CTS, Tel Aviv, Israel, “NC7323XLF”). The field was about 0.25 ha in size. The experiment was performed in the format of 4–5 random blocks per treatment (and control), as we will specify for each year. Each block included five garden beds (repetitions). The garden beds were 6 m wide, 9 m long, and had two rows with 96 cm row spacing.

#### 2.3.1. Sowing and Irrigation Regime

The experimental plots were sown to a depth of 4 cm with seven plants per meter and germinated the following day using a frontal irrigation system. Watering was carried out by dripline irrigation and was controlled by a computerized irrigation system, at a flow rate of 0.6 l h^−1^, using a 20 mm drip irrigation line for each row (Dripnet PC1613 F, Netafim USA, Fresno, CA, USA). The watering regime was 3 cubic meters per 0.1 ha per day. The irrigation was applied every two days, and the total amount of water supplied to the field during the whole season was approximately 400 mm per season. According to the Consultation Service’s (SAHAM, Israel Ministry of Agriculture) recommended growth protocol, all the plants received fertilization and insecticides at the standard dosages.

#### 2.3.2. Complementary Infection Method

The infection method comprised additional complementary steps to enhance and equalize the soil infestation with the pathogen. The plant growth methodology and inoculation were similar to [10,17]. To that end, the experimental plots (except for the negative control) were deliberately infected by infiltrating three sterilized *M. maydis*-inoculated wheat grains into each maize seed with the sowing. These grains were preincubated in a 0.5 L plastic container for 1–3 weeks (until the fungus developed) at 28 °C in the dark with 20 *M. maydis* culture agar disks (prepared as in Section 2.1) per 150 g seeds. The sterilized infected wheat grains were inoculated with a mixture of three *M. maydis* isolates. The isolates selected for these experiments differed for the study years. The 2019 experiment’s inoculation mixture included *Hm5*, *Hm7,* and *Hm21*, while the 2020 experiment’s blend included *Hm2*, *Hm29,* and *Hm30*.

A naturally infested field is populated with pathogen lines having different aggressiveness, as reported in Spain [13,33], Egypt [14], and Israel [6,28]. Such a mixture of pathogen lines results in a more severe disease burst than by using a single line. The Gadash experimental farm that was chosen for the experiments was less heavily infected. Therefore, we deliberately infected it with a mixture of the pathogen’s lines (most from nearby commercial fields in the Hula Valley, see Table 1) pre-tested in a preliminary experiment and which had resulted in a more severe infection than a sole pathogen isolate. In the subsequent year (2020), we chose to enrich the already infected soil with new lines of the pathogen (also from nearby commercial fields, Table 1) to intensify the disease and mimic the natural populations that including several lines.

#### 2.3.3. Trichoderma-Based Biocontrol Treatments

In both years, the *Trichoderma* biocontrol treatment was carried out by adding three sterilized wheat grains enriched with leading biocontrol *Trichoderma* spp. candidates to each seed with sowing. The wheat grains were enriched after they were incubated in DDW (during the night) and sterilized. To 150 g grains, 20 *M. maydis* culture agar disks were added. The disks were 6 mm in diameter and were cut from colonies grown as described in Section 2.2. The enriched grains were incubated at 28 °C in the dark for about 10 days (or until the fungus developed). The 2019 experiment included the species and *T. longibrachiatum* (T7407) and *T. asperelloides* (T203). The 2020 repetition had *T. asperellum* (isolate P1) treatment instead of *T. asperelloides* (T203) treatment, which was less efficient in the 2019 experiment, as detailed in the Results section (Table 1).

### 2.4. Trichoderma spp. Late Wilt Control in the 2019 Growing Season

The first field experiment was performed in the summer and autumn of (6 August–30 October) 2019, a growth period of 85 days. Plants emerged above the ground surface six days after planting, and the emergence evaluation was carried out one day later. Plants were first pollinated when they reached 70% silk on 29 September 2019 (54 days after sowing, DAS). The pollination continued for one week. Most experimental plants showed dehydration symptoms at a different degree of severity at the experiment’s end. From each experimental group (treatment), 10 representative plants were analyzed for their development values and health status, and the first internode’s pathogen DNA relative accumulation was determined. Health assessment was carried out for the whole plant and was based on four categories: healthy (1), minor symptoms (2), dehydrated (3), dead (4).

### 2.5. Trichoderma spp. Late Wilt Control in the 2020 Growing Season

#### 2.5.1. Enhancing the Disease Using Wooden Toothpicks Inoculation

The 2020 experiment was performed on the same experimental Gadash farm in the late summer and autumn of (10 September–1 December) 2020, a growth period of 82 days. This subsequent experiment was conducted in the autumn for practical reasons (mainly the availability of the field). Thus, the conditions were less favorable for inducing LWD. Therefore, to maximize disease burst, two garden beds (out of five) in each repeat were inoculated once more, 20 days from sowing. This inoculation was conducted by stabbing the lower stem section near the ground once with a wooden toothpick dipped in *M. maydis* mycelium (Figure 1). The mycelium was prepared by combined growth in a liquid PDB of the isolates: Hm2, Hm7, Hm30 (10 fungal discs from each isolate in 150 mL PDB). These isolates were grown (in a mixture) at 28 ± 1 °C for one week. Sterilized toothpicks were added to the isolates’ mix 24 h before being used to inoculate the plants. Each stalk was inoculated once by stabbing and leaving the infected wooden toothpick stuck in the stem. The control plots were wounded with clean, sterilized toothpicks.

#### 2.5.2. The 2020 Experiment

Plants emerged above the ground surface ca. one week after sowing and sprouts’ emergence evaluation was carried out 11 DAS. Pollination occurred on 29 October 2020 (49 DAS). The pollination phase followed the male flowering and the beginning of the silk (the petioles of the female part) appearance, and the beginning of their bending. Most plants had a healthy and vital appearance, normal development, and yield (detailed in the Results) at the season’s end. Still, close inspection of the cobs and lower stem (first aboveground internode) revealed disease symptoms, which were analyzed using four categories: healthy (1), minor symptoms (2), moderate symptoms (3), diseased (4). Growth parameters (aboveground parts fresh weight, number of leaves, and height), yield determination, and molecular diagnostic (to track the pathogen DNA) were measured in five plants selected arbitrarily from the center of each of the experiment’s garden beds. The average of these five plants was calculated and determined as one repeat. Each experimental group (controls and biological intervention, with or without stabbing) included eight repetitions.

### 2.6. qPCR Diagnosis of M. maydis DNA in the Maize Plants

qPCR was performed on the plants’ roots at the sprouting phase (40 DAS) and the first aboveground internode at the season’s end (82–85 DAS). The plant tissues were washed twice thoroughly with tap water with sterile DDW and cut into ca. 2 cm sections. Each repetition’s weight was adjusted to 0.7 g. DNA isolation and purification were carried out according to Murray and Thompson’s (1980) protocol [34], with slight modifications, as previously described [28].

The DNA samples were stored at −20 °C and used for qPCR, as previously described [17]. This molecular method is based on the qPCR protocol [35] that was optimized to detect the DNA of *M. maydis* using A200a species-specific primers [36,37] (sequences in Table 3). The housekeeping gene used to normalize the pathogen DNA was COX, a gene encoding the mitochondria’s last enzyme in the cellular respiratory electron transport chain—cytochrome c oxidase [38]. This gene’s amplification was carried out using the COX F/R primer set (Table 3). The ΔCt model was used to calculate the relative gene abundance [39]. All amplifications were performed in triplicate, and similar efficacy was assumed.

The qPCR reactions were conducted using the Sequence Detection System ABI PRISM 7900 HT (Applied Biosystems, Foster City, CA, USA) and 384-well plates. The qPCR settings were as follows: 5 µL total reaction volume per sample well that include 2 µL of DNA sample, 2.5 µL of iTaq™ Universal SYBR Green Supermix (Bio-Rad Laboratories Ltd., Hercules, CA, USA), and 0.25 µL of each of the primers (forward and reverse, 10 µM from each primer per well). The qPCR cycle program involved the precycle activation phase (1 min at 95 °C), denaturation (15 s at 95 °C) for 40 cycles, annealing and extension (30 s at 60 °C), and finalized by a melting curve.

### 2.7. Statistical Analysis

A randomized statistical design was used in the full-growth season experiments. All data presented were subjected to a statistical analysis using the JMP program, 15th edition (SAS Institute Inc., Cary, NC, USA). The one-way analysis of variance (ANOVA) was applied with a significance threshold of *p* < 0.05, followed by post hoc of the Student’s *t*-test for each pair (without multiple comparisons correction). Ordinarily, in lengthy field condition experiments, the measurements result in a high level of differences within the results due to variations in the environmental conditions, host susceptibility, and the LWD pathogen’s spreading nature [17,28]. Therefore, relatively high standard error data resulted in most of these tests, and statistically significant differences could hardly be identified. Thus, most of the results obtained were not statistically different in this test, even if the average differences between the results were, on some occasions, twofold. Such differences can sometimes be identified as significant (*p* < 0.05) when a one-tailed *t*-test (which is a more powerful test) compared each treatment separately to the infected control, indicated by an asterisk (*).

## 3. Results

The current study focused on experimentally targeting the maize late wilt disease’s development and its damages through biological *Trichoderma*-based control. To this end, field experiments conducted during two growing seasons explored the potential of *T. asperelloides* (T203), *T. longibrachiatum* (T7407), and *T. asperellum* (P1) as bioprotective treatments against LWD. These species excelled in preliminary assays conducted previously under controlled conditions [26,27]. They were applied here in the field by directly adding them (in the form of sterilized *Trichoderma*-enriched wheat seeds) to each maize grain with the sowing.

### 3.1. Trichoderma spp. Late Wilt Control in the 2019 Growing Season

In the first field experiment conducted in the summer of 2019, the addition of *T. longibrachiatum* (T7407) prevented the inhibitory influence of the pathogen on the growth measures (root and shoot fresh weight and aboveground parts’ height) at the season’s end (85 DAS, Table 4). These parameters were recovered to the degree of the uninoculated control plant and even higher (the plants’ height). A 12% improvement in the plants’ shoot wet biomass and a 6% increase in the plants’ height were recorded compared to the unprotected infected control. However, these differences were not statistically different due to high variations in the results, typical of lengthy open-air experiments. The T7407 treatment had no positive impact on the total cobs’ wet weight and plant health.

The pathogen DNA levels (measured in the first aboveground internode) were low in the treatments. The range of relative DNA levels measured using the qPCR method is 10–1 × 10^−6^ *Mm*/*Cox* ratio. At low DNA levels (usually less than 1 × 10^−3^), pathogen DNA variations are not always reflected in the plants’ growth parameters and disease symptom severity. Indeed, the T7407 treatment’s pathogen DNA levels were higher than in the unprotected plots.

The T203, which had excelled in a previous study [27] that was carried out in pots (in 2018), was ineffective. Adding this species directly to the field soil (in the form of sterilized and enriched wheat grains) neither improved the plant growth parameters nor reduced the pathogen DNA (Table 4). Therefore, this species was not included in the experiment conducted in the subsequent year. Instead, a new *Trichoderma* candidate, *T. asperellum* (P1), an endophyte isolated in 2019 from maize grains of the LWD-sensitive Prelude cv. [26], was tested in the 2020 field experiment.

### 3.2. Trichoderma spp. Late Wilt Control in the 2020 Growing Season

#### 3.2.1. Enhancing the Disease Using Wooden Toothpicks Inoculation

The growing season (autumn) during which the second field experiment was conducted was less optimal for LWD development due to the drop in temperature and rain (Table 2), a known factor in reducing LWD [41]. Furthermore, we had deliberately infected the experimental field for the second year in a row. Therefore, it did not have a long history of LWD, which characterizes some commercial fields in the area (Hula Valley, northern Israel). These two factors raised concerns that the disease may not break out in a way that would allow us to examine the effect of the treatments. Therefore, to increase the chance of getting the disease symptoms, the plots were split, and inoculation using wooden toothpicks was applied to two (out of five) garden beds 20 days after sowing. While the biological treatment carried out was near the seeds in the soil and the stabbing inoculation was carried out in the stem, it should be noted that *T. asperellum* (P1) is an endophyte that probably inhabits the plant stems as well.

#### 3.2.2. The 2020 Experiment Results at the Sprouting Growth Phase

The field examinations included a sprouts’ emergence test on day 11 of growth, sampling to estimate plant development, pathogen infestation on days 41 and 82 of growth, crop evaluation, and symptoms assessment at the season’s end. The aboveground germinating percentages measured were 81 ± 3.3 and 82 ± 5.5 in the plots without the complementary inoculation and in the inoculated plots, respectively. These emergence percentages were higher (without a statistical difference) in the biological treatments: 84 ± 3.4 in the P1 treatment and 87 ± 1.9 in the T7407 treatment. The emergence parameters were not affected by the stabbing inoculation that was conducted nine days later. At the sprouting phase (41 DAS), the *Trichoderma* treatments slightly shielded the growth indices (wet weight and phenological development expressed as the number of leaves) with an albeit more pronounced measurable effect on the plant height (Table 5). These parameters were not affected significantly under the stabbing inoculation influence.

Examination of pathogen DNA levels in the plant roots on day 41 in the plots without the stabbing inoculation showed no significant differences between the treatments or between the treatments and the controls (Table 5). The stabbing inoculation conducted 20 DAS increased the *M. maydis* DNA content of the infected control 2.2-fold, and the P1 and T7407 bioprotective treatments 17-fold and 43-fold, respectively.

#### 3.2.3. The 2020 Experiment Results at the Harvest

In monitoring the plants at the season’s end (82 DAS), most plants had a healthy and vital appearance, normal development, and yield (Table 6). The yield was relatively lower than the typical yield in this cultivar in commercial fields in this area (usually above 2 kg/m^2^) [16]. This result is probably due to the late growing season (autumn). Infecting the plots (the control+ treatment without the stabbing inoculation) led to a 5% reduction (not statistically significant) in yield compared to the non-inoculated plots (the control- treatment). Furthermore, the total yield outcome was not affected by the biological treatments. Still, the T7407 treatment led to a 7% increase in A-class cob (cob weight exceeding 250 g) yield.

The yield indices (Table 6) prove that the stabbing inoculation did not harm the yields compared to the stabbing-free treatments. A similar measure of crops (and even slightly higher in the stabbing inoculation plots) was obtained in all treatments (without statistical significance). It seems that although over 50% of the plants showed lower stem (Figure 2, Figure 3 and Figure 4) and cob (Figure 5 and Figure 6) symptoms of dehydration (most of them mild, grade 2), this did not affect the yield.

#### 3.2.4. Lower Stem Symptoms Evaluation in the 2020 Growing Season

Whereas overall, no prominent dehydration symptoms were observed, close inspection of the lower stem (first aboveground internode) revealed measurable disease symptoms (Figure 2). The lower stem symptoms were minor at the sprouting phase, but the difference between the infected control and the biological treatments was clear. A 67% and 63% increase in the healthy plants was measured in the P1 and T7407 plots, respectively (from 57% healthy plants in the infected control to 95% and 93%, Figure 3). The stabbing inoculation reduced this beneficial impact to 18% and 57% in the P1 and T7407 plots, respectively. This complementary infection also resulted in the first appearance of 6% and 7% of plants with moderate disease symptoms (grade 3) in the uninfected control and the infected control, respectively. In the P1 treatment, the stabbed plants with near-ground stalk portion mild symptoms reached 17%.

Nearly 40 days later (82 DAS), the lower stem symptoms became more severe (Figure 4). The *Trichoderma* spp.-treated plots had 42–44% healthy plants compared to the infected control that had only 22% non-symptomatic plants. A significant difference was found in the T7404 treatment, *p* < 0.05, using the one-tailed t-test (a more powerful assay than the ANOVA, which resulted in no identified differences). This overall healthy plants’ reduction was accompanied by a sharp elevation in the percentages of moderately diseased plants (grade 3) in all experimental groups and first appearance (1–3%) of severely diseased plants (grade 4) in those plots.

#### 3.2.5. Cobs’ Spathes Symptoms Evaluation in the 2020 Growing Season

The disease severity was also assessed at the season-ending according to the dehydration symptoms on the large bracts surrounding the cobs (the spathes, Figure 5). While the percentage of healthy plants (without dehydration symptoms on the cobs’ spathes) was 40% in the infected control, the bioprotective T740 and P1 treatments improved the rate of healthy plants by 14% and 11%, respectively (Figure 6). Even though this improvement was not statistically significant, it was supported by the pathogen DNA evaluation.

#### 3.2.6. *M. maydis* DNA Evaluation at the 2020 Season’s End

On the 82nd day of growth, 3.2- and 4.9-fold lower levels of the pathogen’s relative DNA were measured in the plant tissues treated with T740 and P1, respectively, compared to the infected control (Figure 7). In the wood toothpicks infected plants, the infection area was infested with the fungus, but the plants did not become diseased (Figure 1). Indeed, in monitoring the cobs’ spathes symptoms (Figure 6) and the amount of fungal DNA in the first aboveground internode (Figure 7), no effect was observed for stabbing in the control treatments. Contrary to expectations, the stabbed-infected control had seven-fold lower *M. maydis* DNA levels (with no statistical difference). At the same time, the stabbing treatment seems to have eliminated the advantage of the *Trichoderma* spp. treatments.

## 4. Discussion

The annual global production of maize is increasing at a rate of 1.6%. This rate will not meet the worldwide demands predicted for 2050 [42]. Maize growth suffers from 130 pests and about 110 diseases caused by bacteria, fungi, and viruses globally [43]. Among the diseases, late wilt disease (LWD) caused by *M. maydis* is one of the most severe in an infected area. Biological control against the causal agent of LWD in corn is of great importance, and efforts are being made on the subject around the world.

The current study proposes the use of *Trichoderma* species, *T. asperelloides* (T203), *T. longibrachiatum* (T7407), and *T. asperellum* (P1), against the pathogen under field conditions. All three species had already been tested in the lab and sprouts against the LWD pathogen. They were chosen for this study from our eight *Trichoderma* isolates [27] and 10 endophytes [26] libraries based on their high success in restricting the pathogen. T203 and T7407 also excelled in potted plants over an entire growth period under field conditions [27]. Here, in a commercial field simulation, the bioprotective agents were tested throughout a whole growing season in 2019 and 2020.

The most successful bio-shield treatment was *T. longibrachiatum.* This bioprotective treatment successfully recovered the plants’ sprouts (41 days old) growth values and protected their health during the entire season (82 DAS). It improved growth and yield indices to the level of uninfected control plants and reduced the amount of pathogen DNA in the plants’ tissues by 32%.

Environmental conditions play a pivotal role in the burst and harshness of plant diseases in the familiar phytopathology triangle model, including also the pathogen and the host. The 2020 experiment was conducted in autumn for practical reasons (mainly the availability of the field). This season was, as expected, colder and rainier, and these conditions were probably the cause of the low yields and a minor outbreak of LWD. Indeed, early sowing of corn in Egypt reduced LWD [44], while late summer planting reduced disease severity in India [45].

While that situation may not be ideal for studying preventive treatments, it provides us with a unique opportunity to explore the pathogenesis under conditions that do not allow the appearance of prominent symptoms. Indeed, the plants seemed healthy and vivid in all plots. Nonetheless, close inspection of the lower stem and the cobs’ spathes revealed a different picture, with up to 80% and 60% symptomatic plants, respectively. Tracking the pathogen DNA inside the plants’ roots support these data, with proximity between the DNA levels and symptoms evaluation. Following *M. maydis* DNA could provide additional important information. For example, the stalk-stabbing inoculation bypassed the protection afforded by the *Trichoderma* treatments in the soil, resulting in a high increase in fungal DNA in those treatments. The symptoms’ evaluation supported this conclusion.

For a deeper understanding of the environmental role, we compared the meteorological data and maize LWD in northern Israel (Hula Valley in Upper Galilee) in the growing seasons of 2016–2020 (Table 7). This comparison is possible since data were collected from one field (Amir) or nearby areas (located ca. 10 km from it). Moreover, the same maize cultivar (Prelude, LWD-susceptible sweet maize) was tested, and the same assessment methods were used (qPCR of the lower stem section and cobs’ spathes dehydration symptoms’ evaluation). This summarization strengthens the possible connection between temperature and precipitation, symptom severity, and pathogen DNA in the stalk. The data suggest that the disease burst is most harsh when temperatures climb to 27 degrees and above. The disease breaks out slightly at temperatures below 26 degrees, with almost no noticeable symptoms (see spring-summer 2018 and autumn 2020). However, these preliminary data require additional support, which could be provided from future studies. In addition, the conclusion about the temperature threshold for the disease burst is probably an oversimplification since other factors are most likely involved.

Interestingly, while a severe disease outbreak was accompanied by a sharp elevation in fungal DNA inside the stem (spring–summer 2016, 2017, and summer 2018), when the field was severely affected and collapsed with 100% dry plants, those DNA levels dropped. We assumed that when the maize host tissues dry out towards the end of the growth period, the fungus goes into the asexual reproduction phase and develops sclerotial bodies and spores [46]. At that time, the primary hyphae biomass gradually comes apart. This hypothesis regarding the connection between symptom development and DNA quantity should be studied more deeply in follow-up studies.

Low water potential is considered one of the most influential factors enhancing LWD progression [47,48]. *M. maydis* is sensitive to low oxygen conditions in wet soils [48]. In contrast, a high oxygen atmosphere promotes the pathogen’s colonies’ growth [29]. Reviewing the literature concluded that frequent watering or saturated soils reduced late wilt by influencing the plant’s surrounding resistance to the pathogen and the plant’s immunity. The soil microorganisms’ communities that antagonize *M. maydis* may be influenced by excessive moisture conditions. Indeed, water availability may be the most central environmental factor affecting the soil’s microbial community and activities [29]. Floods may increase anaerobic conditions that stimulate lytic microorganisms to degrade the pathogen’s sclerotia and reduce its survival potential. There are several supporting pieces of evidence for this line of thinking*. M. maydis* is considered a poor competitive saprophyte compared to other microorganisms in the soil [7]. Moreover, corn did not develop late wilt following paddy-cultivated rice, which increases the availability of Mn for subsequent crops.

There is also supporting data regarding the influence of high water-potential on a plant’s immunity to LWD. Moisture stress is a major influencing factor for a plant’s ability to cope with the disease [47]. Twenty-one-day-old maize plants subjected to irrigation to field capacity were healthy and had higher transpiration rates and relative water contents than water-stressed diseased plants. The LWD pathogen infection resulted in a reduction in the number of vascular bundles in the cross-section of the internode. Xylem vessels’ occlusion may be the most important factor causing the disease symptoms. Therefore, the values of the phloem area per unit leaf area are crucial. Indeed, these values decreased significantly in water-stressed and infected plants [47]. In healthy non-stressed maize plants, the high number of vascular bundles in the internodes and the greater phloem area per leaf area compared with diseased plants may contribute to a faster translocation rate and LWD resistance.

The vastly studied *Trichoderma* isolate, *T. asperelloides* (T203), surpasses the 2018 field condition pot experiment conducted in our previous study [27] when applied to each seed with the sowing. However, in contrast to *T. longibrachiatum* (T7407) and *T. asperellum* (P1), this treatment was ineffective (in the current study) when added to the field soil in the form of sterilized and enriched wheat grains. Subsequent studies should address this issue and verify that the application method did not lead to this unsuccessful result.

Future green applications based on *T. longibrachiatum* (T7407) and *T. asperellum* (P1) as a bioprotective shield against *M. maydis* may benefit if the two species are applied together. T7404 was obtained from a marine source (the Mediterranean sponge *Psammocinia* sp.) [36]. Such marine *Trichoderma* isolates may be used in fields irrigated with relatively high salinity water. It was recently suggested that promoting rhizobacteria seed-coated treatments reduced LWD to 82.8% in regular soil and 79.3% in saline soil in the field [29]. This treatment also improved seed germination percentage and plant growth indices. Thus, T7407 may be combined with promoting rhizobacteria in future treatment. Moreover, *T. asperellum* alleviated alkaline stress in the saline-alkaline sensitive maize variety [29] by promoting photosynthesis to supply energy and more raw materials for nitrogen metabolism, thus improving nitrogen metabolism and the capacity for material production in maize seedlings.

*T. asperellum* (P1) is an endophyte isolated in our laboratory from corn seeds of a maize cultivar susceptible to LWD [26]. As such, it most likely maintains a symbiotic lifestyle within the host plant tissues. It may provide ongoing protection throughout the growing season. Indeed, *Trichoderma* species are able to induce maize plants resistance systemically against diseases through the activation of salicylic acid (SA), jasmonic acid (JA), brassinolide (BR), reactive oxygen species (ROS), and defense enzymes [49].

This is the first examination of *T. asperellum* against *M. maydis* in the field to the best of our knowledge. A similar study was carried out with the same species against other maize pathogens. Several examples are presented here. *T. asperellum* isolated from an African maize line significantly inhibited the growth of *Fusarium verticillioides* in an in vitro competition assay [50]. This species’ GDFS1009 granules achieved a control efficiency of 49.67% against *Fusarium graminearum* under field conditions [51]. When the treatment was applied for three consecutive years, it caused significant control of stalk rot and increased yields. In another study, the combination of *T. asperellum* with betel extract was effective against the maize downy mildew pathogen (*Peronosclerospora* sp.) [52]. A synergistic effect in this combination resulted in reducing the disease incidence. These samples imply that field biocontrol application of both *T. longibrachiatum* (T7407) and *T. asperellum* (P1) may also offer a solution for diseases other than late wilt. If adequately developed into final products and combined with other control methods, the *Trichoderma*-based control could play an essential role in maize crop protection against late wilt.

An integrated LWD control strategy can combine the *Trichoderma* species inspected here with other environmentally friendly solutions such as biopesticide that are gaining increasing importance [53]. In India, for instance, microbial biopesticides research is evolving rapidly, and such pesticides comprise ≈ 5% of the market [54]. Biopesticides developed from pathogenic viruses, bacteria, fungi, nematodes, and plants’ secondary metabolites, are alternatives to chemical pesticides and are a significant component of many pest control programs [55]. New bacteria-based biopesticides are being constantly developed. For example, it was shown that volatile compounds and peptides from the bacteria *Bacillus subtilis*, *Staphylococcus aureus*, and *Pseudomonas aeruginosa* inhibited the hyphal growth and melanin production of *A. solani* [56].

## 5. Conclusions

Biological pesticides against plant diseases in general and the cause of late wilt disease (LWD) in corn, in particular, are at the forefront of scientific research around the world. The LWD agent, *Magnaporthiopsis maydis*, is most devastating to cornfields in highly infected areas such as Israel, Egypt, and Spain. Restricting the disease is an urgent need, and green control methods are highly requested. The current study focuses on the ability of three *Trichoderma* isolates to limit the pathogen in the field during two consecutive years. *T. longibrachiatum* excels as a bioprotective treatment. It successfully recovered the plants’ sprouts (41 days old) growth values and protected their health during the entire season (82 DAS). This biocontrol agent-based treatment improved growth and yield indices to the level of the uninfected control plants and reduced the amount of pathogen DNA in the plants’ tissues by 32%. The present study improves our understanding and coping with LWD and is another step towards developing an environmentally-friendly pesticide solution against it. Future applications should evaluate a combined treatment that includes two or all three *Trichoderma* species inspected in this study since each species brings some unique advantages to this end. It would also be worthwhile to test the *Trichoderma* species’ combination with other green solutions such as promoting rhizobacteria. Finally, under some circumstances, such as severe cases, an integrated control method that combines *Trichoderma*-based biocontrol and chemical fungicides may be the best solution. Such a solution would require pre-testing the effect of these chemical fungicides on the *Trichoderma* species since fungicides could reduce their impact. If adequately developed, such an integrative control method’s main benefit would be reducing the use of pesticides.

## Figures and Tables

**Figure 1 jof-07-00444-f001:**
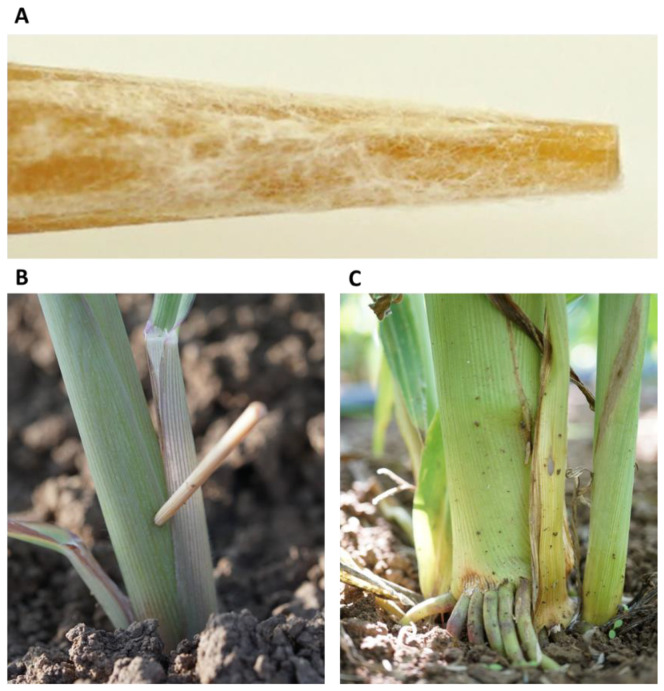
Field plants’ inoculation by stabbing with wooden toothpicks. This procedure was applied in the Gadash farm 2020 experiment. (**A**). Closeup photograph of the toothpick tip after 24 h of inoculation in the *Magnaporthiopsis mayd*is isolates’ mix. (**B**). Field inoculation of 20-day-old sprouts by toothpick stabbing at the aboveground portion of the stem. (**C**). The lower stem portion of the stab-inoculated 82-day-old plant. At this age, the wounded area is infected, populated with fungus, and shows disease symptoms, but the plant as a whole was less affected.

**Figure 2 jof-07-00444-f002:**
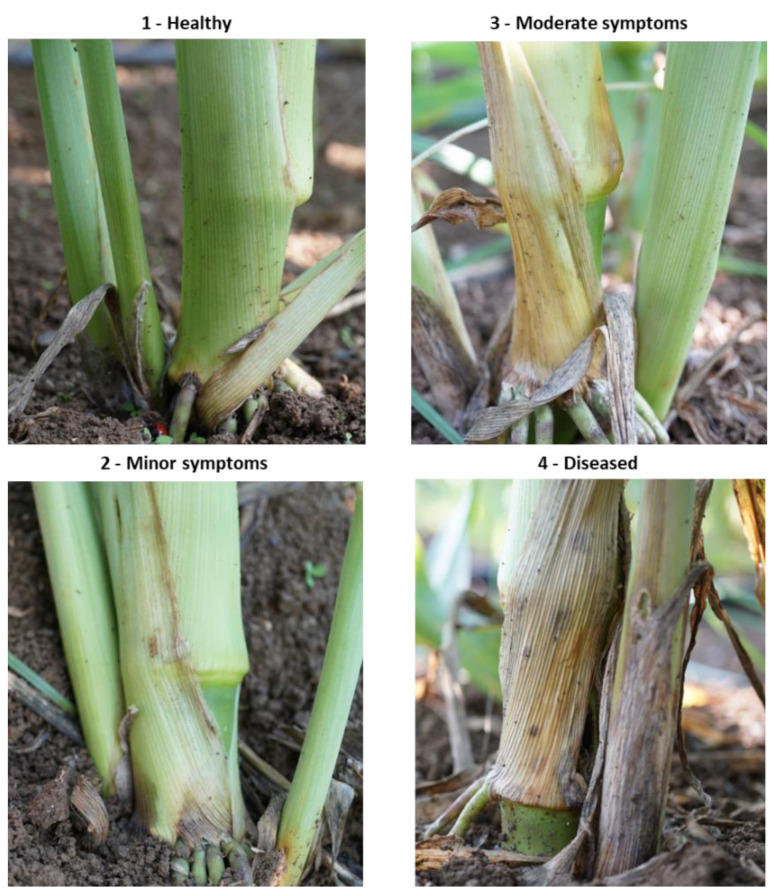
Classification of the lower stem (first aboveground internode) disease symptoms at the end of the 2020 field experiment (82 DAS). Representative images that are showing increasing degrees of late wilt symptoms. Symptoms were analyzed using four categories: healthy (1), minor symptoms (2), moderate symptoms (3), and diseased (4).

**Figure 3 jof-07-00444-f003:**
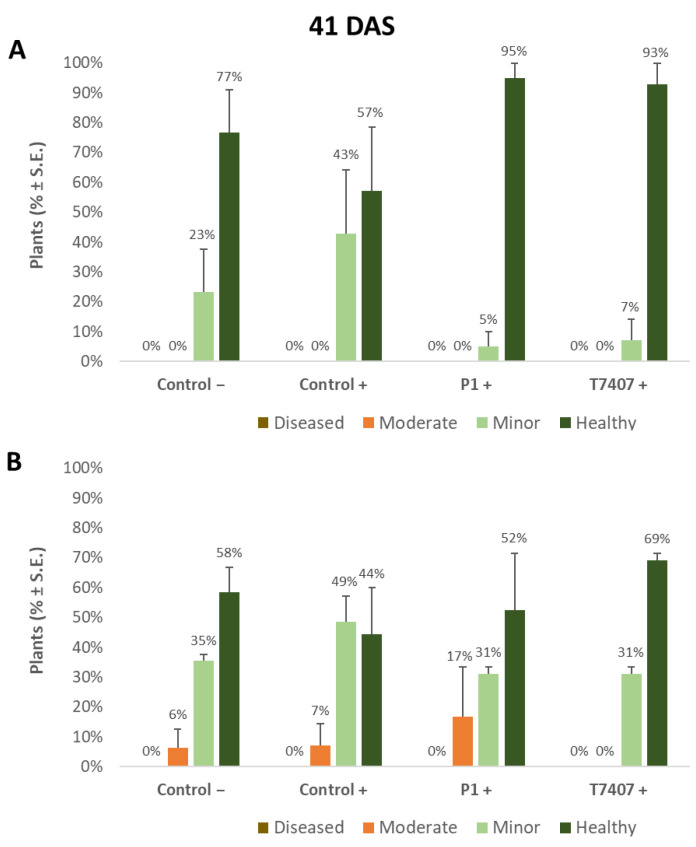
Quantitative evaluation of lower stem disease symptoms in the 2020 field experiment (41 DAS). (**A**). Stabbing-free treatments. (**B**). Stabbing inoculation treatments. Categories are described in Figure 2. Control− are plots without a complementary inoculation. Control+ are plots with *M. maydis* complementary infection. The other treatments are infected plots with *T. asperellum* (P1) and *T. longibrachiatum* (T7407). The control− are stabbing plots wounded with clean, sterilized toothpicks. Values are calculated based on 10–23 plants in each treatment in at least two repeats. Deviation bars represent standard error. No statistical difference between the treatments or the treatments and controls could be identified using the one-way ANOVA test.

**Figure 4 jof-07-00444-f004:**
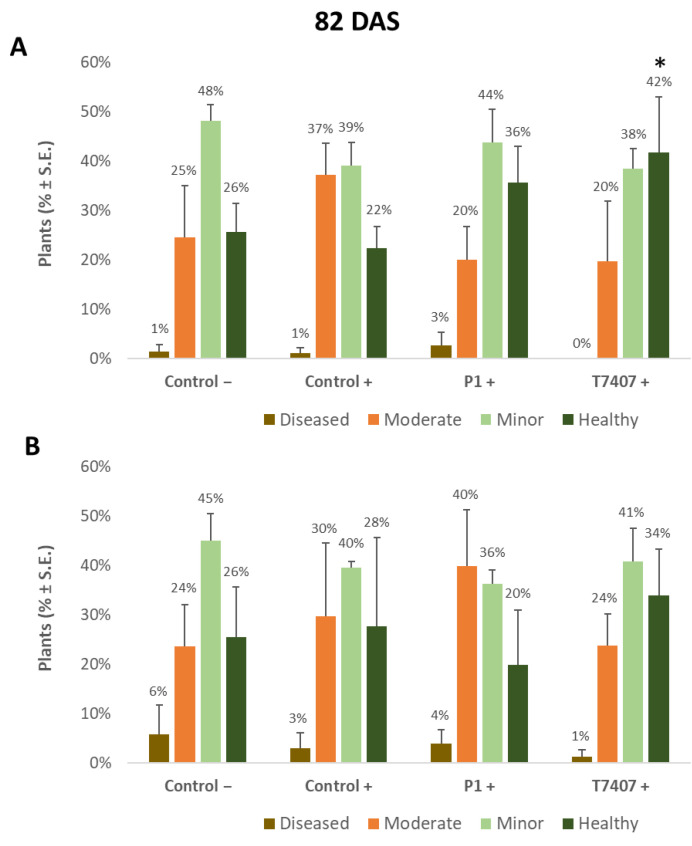
Quantitative evaluation of lower stem disease symptoms at the end of the 2020 field experiment (82 DAS, 33 days after fertilization, DAF). (**A**). Stabbing-free treatments. (**B**). Stabbing inoculation treatments. Categories are described in Figure 2. The experimental treatments and controls are depicted in Figure 3. Values are calculated based on 61–158 plants in each treatment in at least two repeats. Deviation bars represent standard error. No statistical difference between the treatments or the treatments and the controls could be identified using the one-way ANOVA test. Yet, a one-tailed *t*-test compared to the infected control (a more powerful test) revealed significant differences (*p* < 0.05), indicated by an asterisk (*).

**Figure 5 jof-07-00444-f005:**
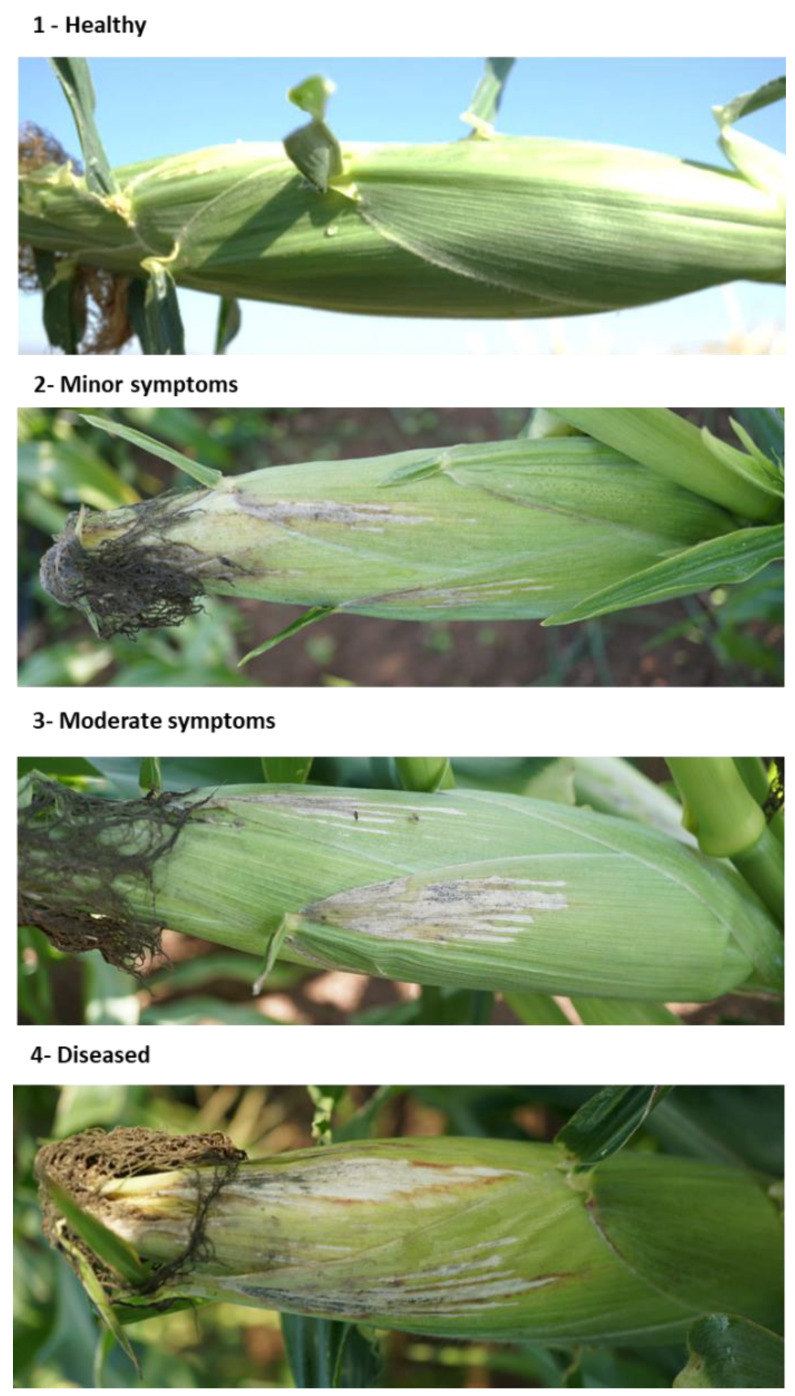
Classification of the cobs’ spathes disease symptoms at the end of the 2020 field experiment (82 DAS). Representative images are in increasing degrees of late wilt symptoms. The dehydration symptoms on the large bracts surrounding the cobs (the spathes) were estimated according to the following scale: 1—healthy with no signs of dehydration, 2—minor symptoms up to 20% of the cob surface, 3—moderate symptoms that cover 30–40% of the cob surface, 4—diseased with 50% or more dehydrated cobs’ area.

**Figure 6 jof-07-00444-f006:**
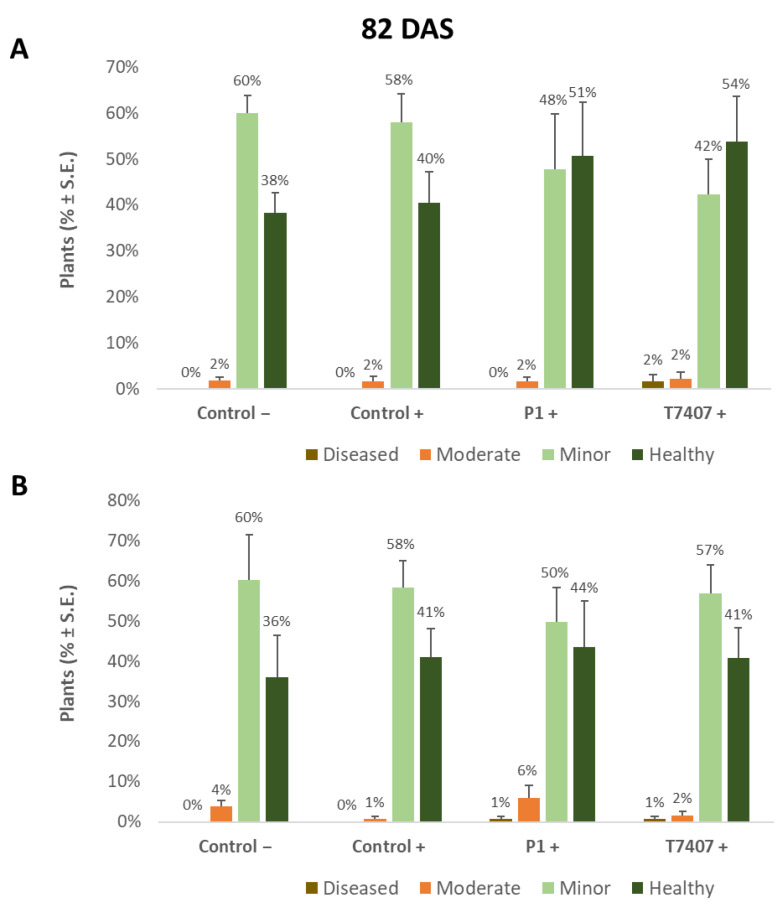
Quantitative estimation of cobs’ spathes symptoms at the end of the 2020 field experiment (82 DAS, 33 DAF). (**A**). Stabbing-free treatments. (**B**). Stabbing inoculation treatments. Categories are described in Figure 5. The experiment’s treatments and controls are depicted in Figure 3. Values are calculated based on 118–241 plants in each treatment in at least four repeats. Deviation bars represent standard error. No statistical difference between the treatments or between the treatments and the controls could be identified using the one-way analysis of variance (ANOVA) test.

**Figure 7 jof-07-00444-f007:**
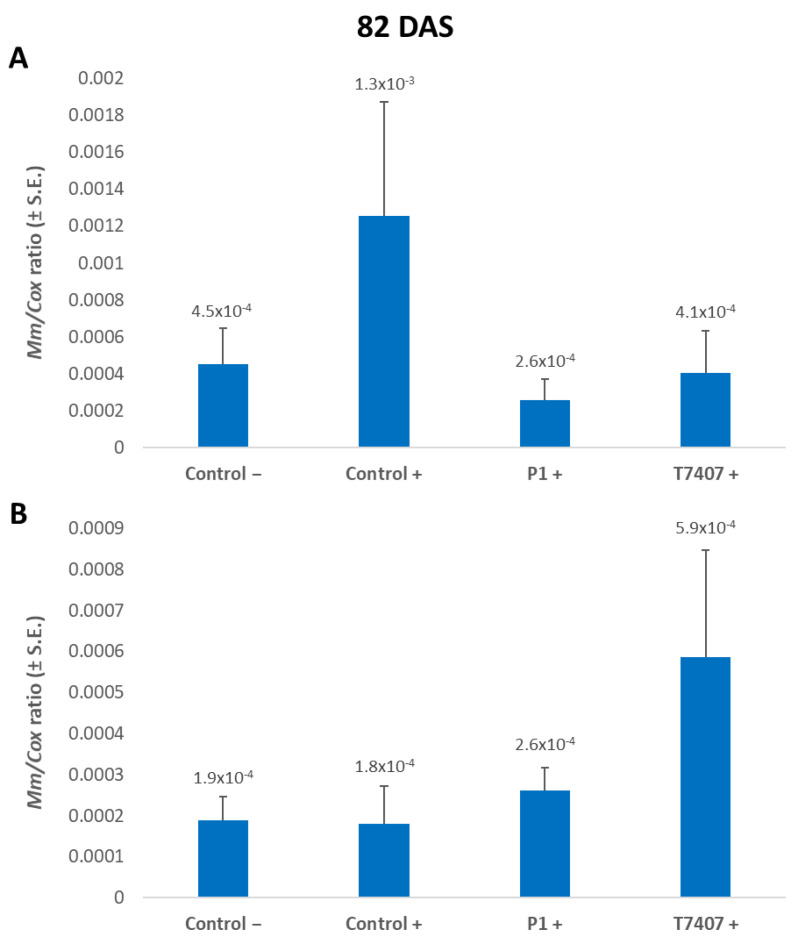
A qPCR estimate of *M. maydis* DNA at the end of the 2020 field experiment. (**A**) Stabbing-free treatments. (**B**) Stabbing inoculation treatments. The relative amount of DNA (*Mm*) of *M. maydis* normalized to the DNA of cytochrome C oxidase (*Cox*) was assessed 82 days after sowing (33 DAF) in the plants’ first aboveground internode. The experiment’s treatments and controls are depicted in Figure 3. Error lines represent a standard error of an average of 10 repetitions (plants). No statistical difference was recognized using the one-way ANOVA test.

**Table 1 jof-07-00444-t001:** Fungi used in this research.

Species	Isolate Designation	Origin	Isolation Location in Northern Israel	Reference	Tested Here in the Field
*Magnaporthiopsis maydis*	Hm5	*Zea mays*, 618 cv.	Sasa field ^2^	[28]	2019
*Magnaporthiopsis maydis*	Hm21	*Zea mays*, Prelude cv.	Amir field ^2^	This study	2019
*Magnaporthiopsis maydis*	Hm7	*Zea mays*, Colossus cv.	Dovrat field ^3^	[28]	2019, 2020
*Magnaporthiopsis maydis*	Hm2	*Zea mays*, Jubilee cv., CBS 133165	Sede Nehemia field ^2^	[6,28]	2020
*Magnaporthiopsis maydis*	Hm29	*Zea mays*, 32D99 cv.	Malcia field ^2^	This study	2020
*Magnaporthiopsis maydis*	Hm30	*Zea mays*, 32D99 cv.	Malcia field ^2^	This study	2020
*Trichoderma asperelloides*	T203	ATCC 36042, CBS 396.92		[27,32]	2019
*Trichoderma longibrachiatum*	T7407	*Psammocinia* sp. ^1^		[25,27]	2019, 2020
*Trichoderma asperellum*	P1	*Zea mays* (Prelude cv.)		[26]	2020

^1^ Mediterranean sponge, ^2^ Upper Galilee Regional Council, Hula Valley. ^3^ Jezreel Valley Regional Council.

**Table 2 jof-07-00444-t002:** Meteorological data for the 2019 and 2020 experiments ^1^.

Parameters	2019	2020
Dates	6 August–30 October	10 September–1 December
Temperature (°C)	25.8 ± 5.7	23.0 ± 7.1
Humidity (%)	63.2 ± 19.2	64.0 ± 23.4
Soil temp. top 5 cm (°C)	30.7 ± 11.2	25.6 ± 7.6
Radiation (W/m^2^)	235.3	172.2
Precipitation (mm)	53.3	140.6
Evaporation (mm)	556.58	299.2

^1^ Data (average ± standard deviation) according to Israel Northern Research and Development (Hava 1 meteorological station) data.

**Table 3 jof-07-00444-t003:** Primers used in this study.

Pairs	Primer	Sequence	Uses	Amplification	References
Pair 2	A200a-for A200a-rev	5′-CCGACGCCTAAAATACAGGA-3′ 5′-GGGCTTTTTAGGGCCTTTTT-3′	qPCR target gene	200 bp *M. maydis* species-specific fragment	[6]
Pair 3	COX-F COX-R	5′-GTATGCCACGTCGCATTCCAGA-3′ 5′-CAACTACGGATATATAAGRRCCRR AACTG-3′	qPCR control	Cytochrome c oxidase (*COX*) gene product	[38,40]

**Table 4 jof-07-00444-t004:** *Trichoderma* spp. late wilt control in the 2019 growing season ^1^.

Growth Parameter	Control −		Control +		T203 +		T7407 +	
	Mean	S.E.	Mean	S.E.	Mean	S.E.	Mean	S.E.
Emergence (plants/m^2)^) 7 DAS	16.1 ^A^	0.19	16.4 ^A^	0.40	15.7 ^B^	0.65	16.0 ^A,B^	0.26
Root weight (g)	43.5	4.5	38.7	3.9	34.4	3.7	42.4	3.9
Shoot weight (g)	240.9	12.2	217.5	16.2	212.8	9.5	242.7	16.8
Shoot height (cm)	169.0	2.9	168.2	4.7	168.8	4.2	177.6	3.1
Cob wet weight (g)	245.8	17.1	261.2	19.5	242.8	31.5	210.5	25.2
Health index (1–4	1.2	0.2	1.2	0.2	1.3	0.2	1.5	0.2
qPCR (*Mm*/*Cox* ratio)	7.3 × 10^−4^	1.5 × 10^−3^	5.3 × 10^−4^	1.1 × 10^−3^	1.9 × 10^−3^	3.0 × 10^−3^	1.6 × 10^−3^	1.9 × 10^−3^

^1^ Maize average growth indices 85 days after sowing (DAS), 31 days after fertilization (DAF). The experiment was performed in a Prelude corn variety susceptible to late wilt. Control– are plots without a complementary inoculation. Control+ are plots with *M. maydis* complementary infection. The other treatments are infected plots with *Trichoderma asperelloides* (T203) and *T. longibrachiatum* (T7407). Health assessment was made on the whole plant and was based on four categories: healthy (1), minor symptoms (2), dehydrated (3), dead (4). Values represent an average of 10 replications ± standard error. If existing, a statistically significant (*p* < 0.05) difference between the treatments at the same measure is indicated by different letters (^A,B^).

**Table 5 jof-07-00444-t005:** *Trichoderma* spp. late wilt control in the 2020 growing season at 41 DAS ^1^.

Treatment	SI ^2^	Wet Weight (g)		Number of Leaves		Plant Height (cm)		qPCR (*Mm*/*Cox*)	
		Mean	S.E.	Mean	S.E.	Mean	S.E.	Mean	S.E.
Control-	−	255.7	10.6	9.87	0.2	115.4	2.1	5.5 × 10^−4^	3.3 × 10^−4^
Control-	Stabbing	240.2	17.1	9.80	0.3	111.7	4.1	3.1 × 10^−3^	2.4 × 10^−3^
Control+	−	240.5	11.0	9.79	0.2	113.7	2.3	2.5 × 10^−4^	1.8 × 10^−4^
Control+	+	243.3	16.0	9.75	0.2	112.2	3.0	7.2 × 10^−4^	6.6 × 10^−4^
P1	−	245.1	13.6	9.00	0.2	116.9	2.5	3.0 × 10^−4^	1.9 × 10^−4^
P1	+	248.8	16.3	10.50	0.2	118.7	3.6	5.1 × 10^−3^	2.8 × 10^−3^
T7407	−	258.6	15.3	9.90	0.2	119.9	2.3	3.1 × 10^−4^	2.1 × 10^−4^
T7407	+	246.9	9.7	10.08	0.2	118.6	2.3	1.3 × 10^−2^	1.0 × 10^−2^

^1^ The experiment was performed in late wilt-susceptible Prelude cv. Control- are plots without a complementary inoculation. Control+ are plots with *M. maydis* complementary infection. The other treatments are infected plots with *T. asperellum* (P1) and *T. longibrachiatum* (T7407). Values represent an average of five replications ± standard error. ^2^ SI+ are the stabbing inoculation plots. The control- stabbing plots were wounded with clean sterilized toothpicks.

**Table 6 jof-07-00444-t006:** *Trichoderma* spp. late wilt control in the 2020 growing season at the season’s end ^1^.

TREATMENT	SI ^2^	YIELD (KG/M2)		A-CLASS (GR’)		B-CLASS (GR’)	
		Mean	S.E.	Mean	S.E.	Mean	S.E.
CONTROL-	−	1.20	0.08	369.2	14.2	160.3	17.3
CONTROL-	Stabbing	1.11	0.15	345.9	12.9	158.3	5.5
CONTROL+	−	1.14	0.08	341.9	20.3	152.4	14.9
CONTROL+	+	1.19	0.08	347.8	17.2	195.8	33.1
P1	−	1.09	0.05	333.1	20.9	180.2	16.1
P1	+	1.22	0.04	363.3	25.6	187.3	16.9
T7407	−	1.16	0.17	365.5	43.8	167.8	24.4
T7407	+	1.20	0.05	354.5	13.9	156.9	19.2

^1^ The experiment and treatments are described in Table 5. Data were collected at 82 DAS (33 DAF) in five replications ± standard error. The yield classified as A class had a cob weight above 250 g. ^2^ SI, stabbing inoculation.

**Table 7 jof-07-00444-t007:** Environmental conditions and maize late wilt disease in northern Israel in the growth seasons of 2016–2020 ^1^.

Location and Year	Dates	Average Temp.	Precipitation	Dehydration	*M. maydis* DNA	Reference
Amir 2016 (spring–summer)	25 May–2 August (75 DAS)	27 °C	0.6 mm	60% (69 DAS) 100% (75 DAS)	0.05 (60 DAS) 7.8 × 10^−05^ (75 DAS)	[17]
Amir 2017 (spring–summer)	24 May–2 August (70 DAS)	27 °C	0 mm	73%	6.26	[16]
Neot Mordechai 2018 (spring–Summer)	23 April–5 July (73 DAS)	25 °C	30 mm	Less than 10%	6.5 × 10^−05^	[15]
Amir 2018 (summer)	21 June–5 September (71 DAS)	28 °C	3 mm	72%	0.02	[15]
Gadash farm 2019 (summer)	6 August–30 October (85 DAS)	26 °C	53 mm	30%	5.3 × 10^−04^	This study
Gadash farm 2020 (autumn)	10 September–1 December (82 DAS)	23 °C	141 mm	Less than 10%	4.5 × 10^−04^	This study

^1^ The experiments were conducted in the Amir field in the Hula Valley in Upper Galilee, northern Israel, or in nearby fields (located about 10 km from the Amir field). The sweet susceptible maize Prelude cv. was evaluated in all seasons. Meteorological data (averages) were according to Israel Northern Research and Development Meteorological Station data—Hava 1. Dehydration percentage—the plants were classified as wilted when wilt symptoms appeared on the cobs’ spathes. *M. maydis* DNA—lower stem qPCR results of specific *M. maydis* DNA fragment normalized to the cytochrome c oxidase (*COX*).

## Data Availability

The datasets generated during and/or analyzed during the current study are available from the corresponding author on reasonable request.

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
