# Peer review of "Trichoderma longibrachiatum and Trichoderma asperellum Confer Growth Promotion and Protection against Late Wilt Disease in the Field"

_jof, 2021, doi:10.3390/jof7060444_

Round 1

Reviewer 1 Report

The manuscript by Degani et al. reported the protective effect of some Trichoderma isolates against late wilt pathogen under field conditions. The science is fine. However, major revision is necessary to improve the manuscript.

Abstract

The authors wrote (The results of this study suggest a biological Trichoderma- based protective layer that may have important value in mild cases of LWD and an essential role in integration with chemical fungicides in severe cases). I think the authors should test first the effect of these chemical fungicides on both Trichoderma species. Fungicides could reduce and affect Trichoderma also not only the pathogen.

Introduction

The authors wrote (Thus, when applying biocontrol antagonists, they change along with the pathogen variability in a continuous evolutionary arms race). This sentence should be removed because it gives the reader an impression that the changes in biocontrol antagonists are always positive and useful for its biocontrol activity but it could also have a negative impact on the biocontrol activity.

L81/ Recently we have scanned three Trichoderma isolates with inhibitory activity against M. maydis. Unfortunately, the isolates (T. asperelloides (T.203) and T. longibrachiatum (T.7407) were tested against Botrytis cinerea (BO5.10), Rhizoctonia solani (TP6), and Alternaria alternate but not M. maydis. Please amend this sentence in more precise information.

Materials and Methods

I wonder why the authors used a mixture of three isolates in the first season and a different mixture of three isolates in the second. I recommend using the same mixture from the most virulent isolates of M. maydis in both seasons.

The authors deliberately infected the experimental field. It could be better to use an infected field with LWD!!

You think that it was a unique opportunity to explore the pathogenesis under conditions that do not allow the appearance of symptoms in 2020. Actually, I think that it is better all the time to test the efficiency of biocontrol agents under favorable conditions for the pathogen that could draw a good image of disease control.

Results

P7L292 Please check the concentration of DNA.

P8L296 I couldn’t understand the sentence (The T203, 295
which excels in field condition pot experiment conducted in 2018). Please rewrite this sentence.

Discussion

Why did you use your own scale for disease evaluation?

M. maydis is a systemic pathogen (not local infection), so it was better to use a scale showing disease progress in all the infected plants. The symptoms could be severe in some parts while the others aren’t affected or showing mild symptoms. The percentage of symptomatic plant parts is more useful in this respect.

I think also it was better to test and compare the effect of azoxystrobin with Trichoderma under field conditions which will complete your previous study (Degani et al. 2019) and also improve the conclusion of your manuscript.

Please check the English and grammar throughout the text especially the results. Overall, the paper needs to have the writing improved, and this will allow better clarity in your message.

 Author Response

Responses to Reviewer 1’s comments

We thank the reviewer for investing substantial efforts, which are undoubtedly contributing to this manuscript. The remarks and suggestions improved this paper’s scientific soundness and accurateness. Your contribution is greatly appreciated.

Abstract

The authors wrote (The results of this study suggest a biological Trichoderma- based protective layer that may have important value in mild cases of LWD and an essential role in integration with chemical fungicides in severe cases). I think the authors should test first the effect of these chemical fungicides on both Trichoderma species. Fungicides could reduce and affect Trichoderma also not only the pathogen.

The reviewer is correct; this sentence was pointing towards future research directions. The sentence was removed from the Abstract, and this explanation was added to the Conclusions, lines 617-622: “Finally, under some circumstances such as in severe cases, an integrated control method that combines Trichoderma-based biocontrol and chemical fungicides may be the best solution. Such a solution would require pre-testing the effect of these chemical fungicides on the Trichoderma species since fungicides could reduce their impact. If adequately developed, such an integrative control method’s main benefit would be reducing the use of pesticides.”

Introduction

The authors wrote (Thus, when applying biocontrol antagonists, they change along with the pathogen variability in a continuous evolutionary arms race). This sentence should be removed because it gives the reader an impression that the changes in biocontrol antagonists are always positive and useful for its biocontrol activity but it could also have a negative impact on the biocontrol activity.

We agree with this remark, and the sentence was deleted as suggested.

L81/ Recently we have scanned three Trichoderma isolates with inhibitory activity against M. maydis. Unfortunately, the isolates (T. asperelloides (T.203) and T. longibrachiatum (T.7407) were tested against Botrytis cinerea (BO5.10), Rhizoctonia solani (TP6), and Alternaria alternate but not M. maydis. Please amend this sentence in more precise information.

This is a typographical error. The paragraph was corrected and now reads (lines 75-81): “We recently scanned three Trichoderma isolates with inhibitory activity against M. maydis: T. asperelloides (T.203); T. longibrachiatum (T.7407 from marine source [25]); and T. asperellum (P1), an endophyte isolated in our laboratory from corn seeds of a strain susceptible to LWD [26]. These isolates prevented the pathogen’s growth in culture plates, significantly reduced its establishment and development in seedlings’ corn plant tissues, and resulted in significant improvement in growth and crop indices in potted plants under field conditions [27].”

 Materials and Methods

I wonder why the authors used a mixture of three isolates in the first season and a different mixture of three isolates in the second. I recommend using the same mixture from the most virulent isolates of M. maydis in both seasons. The authors deliberately infected the experimental field. It could be better to use an infected field with LWD!! 

The reviewer is correct; this should be better explained. A naturally infested field is populated with pathogen lines having different aggressiveness (see reports from Spain (García-Carneros, Girón, and Molinero-Ruiz 2011; Ortiz-Bustos et al. 2015), Egypt (Zeller et al. 2002) and Israel (Degani, Dor, et al. 2019; Drori et al. 2013)). Such a mixture of pathogen lines results in a more severe disease burst than the use of a single line. After several years in which we conducted experiments in such a heavily infected field (Amir field, located ca. 10 km from the current work location (Degani, Movshowitz, et al. 2019; Degani et al. 2018), the new owner of the field refused to let us conduct further experiments in that field. No other LWD-infected fields were available for the experiments at that time. So, in 2019, we started experimenting at the Gadash experimental farm (Hula Valley, Upper Galilee, northern Israel). This new field was less heavily infected. Therefore, we deliberately infected it with a mixture of the pathogen’s lines (most from nearby commercial fields) that were pre-tested in a preliminary experiment and resulted in a more severe infection than a sole pathogen isolate. In the subsequent year (2020), we chose to enrich the already infected soil with new lines of the pathogen (also from nearby commercial fields) in order to intensify the disease and mimic the natural populations that include several lines.

The following explanation was added to the Materials and Methods (lines 171-180): “A naturally infested field is populated with pathogen lines having different aggressiveness (see reports from Spain [13,34], Egypt [14], and Israel [6,30]). Such a mixture of pathogen lines results in a more severe disease burst than by using a single line. The Gadash experimental farm that was chosen for the experiments was less heavily infected. Therefore, we deliberately infected it with a mixture of the pathogen’s lines (most from nearby commercial fields, see Table 1) pre-tested in a preliminary experiment and which had resulted in a more severe infection than a sole pathogen isolate. In the subsequent year (2020), we chose to enrich the already infected soil with new lines of the pathogen (also from nearby commercial fields, Table 1) to intensify the disease and mimic the natural populations including several lines.”

Also, we added an explanation to the Materials and Methods (lines 206-208): “This subsequent experiment was conducted in the autumn for practical reasons (mainly the availability of the field). Thus, the conditions were less favorable for inducing LWD.”

You think that it was a unique opportunity to explore the pathogenesis under conditions that do not allow the appearance of symptoms in 2020. Actually, I think that it is better all the time to test the efficiency of biocontrol agents under favorable conditions for the pathogen that could draw a good image of disease control.

We agree with the reviewer. Favorable conditions for the disease outbreak are always preferred. Still, conducting field experiments is a complex operation, depending on many factors (the availability of the field, labor, budget, equipment, etc.) and therefore cannot always be performed under ideal conditions. Such practical responses led to the late season 2020 experiment. Still, studying the pathogenesis in such a situation can provide vital information, has happened here, and can contribute to the accumulative scientific knowledge of the disease.

We revised the relevant paragraph in the Discussion to better explain this (lines 486-490): “The 2020 experiment was conducted in autumn for practical reasons (mainly the availability of the field). This season was, as expected, colder and rainier, and these conditions were probably the cause of the low yields and a minor outbreak of LWD. Indeed, early sowing of corn in Egypt reduced LWD [45], while late summer planting reduced disease severity in India [46].”

 Results

 P7L292 Please check the concentration of DNA.

The sentence is correct. Lines 298-300: “At low DNA levels (usually less than 1 x 10-3), pathogen DNA variations were not always reflected in the plants’ growth parameters and disease symptom severity.”

The qPCR molecular tracking of the pathogen DNA inside the host plant tissues is very sensitive and can identify differences in the DNA concentration up to 1 million-fold. The range of the relative DNA levels measured using the qPCR method is 10 – 1x10-6 Mm/Cox ratio.

We added this information to the text (lines 296-298): “The pathogen DNA levels (measured in the first aboveground internode) were low in the treatments. The range of relative DNA levels measured using the qPCR method is 10 – 1x10-6 Mm/Cox ratio.”

P8L296 I couldn’t understand the sentence (The T203, 295
which excels in field condition pot experiment conducted in 2018). Please rewrite this sentence.

The sentence was rewritten to clarify our intention (lines 302-305): “The T203, which had excelled in a previous study [27] that was carried out in pots (in 2018), was ineffective. Adding this species directly to the field soil (in the form of sterilized and enriched wheat grains) neither improved the plant growth parameters nor reduced the pathogen DNA (Table 4).”

Discussion

Why did you use your own scale for disease evaluation? M. maydis is a systemic pathogen (not local infection), so it was better to use a scale showing disease progress in all the infected plants. The symptoms could be severe in some parts while the others aren’t affected or showing mild symptoms. The percentage of symptomatic plant parts is more useful in this respect.

We agree. In severe disease cases, the whole plant is affected, and the dehydration symptoms are spread across the stem, leaves, and kernels. These disease symptoms have been described extensively in our previous publications (see Degani and Cernica, 2014; Degani et al., 2018; Degani, Movshowitz et al., 2019; Drori et al., 2013). The current situation is different. Most of the plants’ parts were non-symptomatic and healthy, and the symptoms appeared only on the lower stem section (first aboveground internode) and on the cobs’ spathes. So, we scanned multiple plants from each treatment (61-158 lower stem and 118-241 cobs’ spathes at the season’s end) and evaluated the percentage of infected plants. As far as we know, such work has never been done before on this pathogen.

I think also it was better to test and compare the effect of azoxystrobin with Trichoderma under field conditions which will complete your previous study (Degani et al. 2019) and also improve the conclusion of your manuscript.

This is true, and no doubt, will contribute to the evaluation of the biological treatments. However, the current study is already vast and includes many significant experimental results that should be presented to encourage further follow-up research. A field trial that will compare the effect of Azoxystrobin with Trichoderma under field conditions (and may also test their combination after pretesting them in the lab) should be the focus of subsequent work. Such a study will require several months to accomplish, and it is worthy of another article since it will produce many new results.

This subject is presented at the end of the Conclusion section (lines 617-622): “Finally, under some circumstances such as in severe cases, an integrated control method that combines Trichoderma-based biocontrol and chemical fungicides may be the best solution. Such a solution would require pre-testing the effect of these chemical fungicides on the Trichoderma species since fungicides could reduce their impact. If adequately developed, such an integrative control method’s main benefit would be reducing the use of pesticides.”

Please check the English and grammar throughout the text, especially the results. Overall, the paper needs to have the writing improved, and this will allow better clarity in your message.

The entire manuscript was edited by a professional English scientific copy editor.

Bibliography

Degani, O., S. Dor, D. Movshowitz, E. Fraidman, O. Rabinovitz, and S. Graph. 2018. Effective chemical protection against the maize late wilt causal agent, Harpophora maydis, in the field. PLoS One, 13:e0208353.

Degani, O., D. Movshowitz, S. Dor, A. Meerson, Y. Goldblat, and O. Rabinovitz. 2019. Evaluating Azoxystrobin seed coating against maize late wilt disease using a sensitive qPCR-based method. Plant Dis, 103:238-248.

Degani, O., Cernica, G. 2014. Diagnosis and control of Harpophora maydis, the cause of late wilt in maize. Advances in Microbiology, 04:94-105.

Degani, O., Dor, S., Movshovitz, D., Rabinovitz, O. 2019. Methods for Studying Magnaporthiopsis maydis, the maize late wilt causal agent. Agronomy, 9:181.

Drori, R., Sharon, A., Goldberg, D., Rabinovitz, Onn., Levy, M., Degani, O. 2013. Molecular diagnosis for Harpophora maydis, the cause of maize late wilt in Israel. Phytopathologia Mediterranea, 52:16-29.

García-Carneros, A.B., Girón, I., Molinero-Ruiz, L. 2011. Aggressiveness of Cephalosporium maydis causing late wilt of maize in Spain. Communications in Agricultural and Applied Biological Sciences, 77:173-179.

Ortiz-Bustos, C.M., Testi, L. García-Carneros A.B., Molinero-Ruiz, L. 2015. Geographic distribution and aggressiveness of Harpophora maydis in the Iberian peninsula, and thermal detection of maize late wilt. European Journal of Plant Pathology, 144:383-397.

Zeller, K.A., Ismael, A.M., El-Assiuty, E.M., Fahmy, Z.M., Bekheet, F.M., Leslie, J.F. 2002. Relative competitiveness and virulence of four clonal lineages of Cephalosporium maydis from Egypt toward greenhouse-grown maize. Plant Dis, 86: 373-78.

Reviewer 2 Report

Manuscript title: Trichoderma longibrachiatum and Trichoderma asperellum as bioprotective treatments against late wilt disease in the field

Title - Title needs to be reframed. ‘Trichoderma longibrachiatum and asperellum confers protection against late wilt…’. having included more than crop protection studies, growth promotion could have also been added to title or in keywords.

Abstract is very long – shall avoid lengthy introduction and concise – shall start with background, scope of research in two sentences, followed by brief methodology, significant results and inference.

Keywords-better sets of keywords shall be chosen – the ones not in manuscript title.

Experimental procedures are very lengthy and shall be rewritten to make it concise and precise. Separate procedures shall bear individual headings.

Results and procedure shall follow a coherent order and under appropriate subheadings.

Discussion is very long and loses the sense of research in middle. Shall be rewritten in a coherent order substantiating with references not old as 2010.

Inspite of a very interesting and promising research work, the presentation of the work in the manuscript, right from abstract to conclusion is dragging; the discussion section makes it hard to perceive a better understanding of the findings. Discussion section shall be attempted again with a clear flow of correlating the need of experiment, result and outcome – correlating with previous similar citations. The extensive work done by the researchers are not carried out properly in the manuscript. The whole manuscript shall be rewritten in a precise and more understanding manner. The manuscript shall also be checked for typographical and formatting errors.

Author Response

Responses to Reviewer 2’s comments

We would like to express our sincere appreciation to the reviewer for essential and helpful advice. The time and effort invested are greatly appreciated and certainly contributed to the manuscript and improved it. Thank you.

Title - Title needs to be reframed. ‘Trichoderma longibrachiatum and asperellum confers protection against late wilt…’. having included more than crop protection studies, growth promotion could have also been added to title or in keywords.

The title was edited as suggested by the reviewer and is now written: “Trichoderma longibrachiatum and Trichoderma asperellum confer growth promotion and protection against late wilt disease in the field.”

Abstract is very long – shall avoid lengthy Introduction and concise – shall start with background, scope of research in two sentences, followed by brief methodology, significant results and inference.

The reviewer is correct; the Abstract was rewritten to make it shorter, and more concise, focused, and clear.

Keywords-better sets of keywords shall be chosen – the ones not in the manuscript title.

The keywords set was edited as suggested and is now include the following terms: biological control; Cephalosporium maydis; crop protection; field assay; fungus; Harpophora maydis; Magnaporthiopsis maydis; maize; real-time PCR

Experimental procedures are very lengthy and shall be rewritten to make it concise and precise. Separate procedures shall bear individual headings.

We agree; the entire Materials and Methods section was reorganized (with new subheadings), edited, and improved. The following changes were incorporated:

  • 1 subheading was updated to: “Origin and growth of Magnaporthiopsis maydis
  • The paragraph in lines 89-105 was shortened and corrected
  • 2 subheading was added: “Origin and growth of the Trichoderma species”
  • 3 subheading updated to: “Overall description of the field experiments”
  • The paragraph in lines 139-149 was shortened and corrected
  • 3.1 subheading was added: “Sowing and irrigation regime”
  • 3.2 subheading was added: “Complementary infection method”
  • The paragraph in lines 161-171 was updated and corrected
  • 3.3 subheading was added: “Trichoderma-based biocontrol treatments”
  • The entire description of the stabbing inoculation (including Figure 1) was moved to Section 2.5
  • The paragraph in lines 194-203 was updated and corrected
  • Section 2.5 (Trichoderma late wilt control in the 2020 growing season) was updated
  • 5.1 subheading was added: “Enhancing the disease using wooden toothpicks inoculation”
  • 5.2 subheading was added: “The 2020 experiment protocol”
  • 6 subheading was updated to: “qPCR diagnosis of M. maydis DNA in the maize plants”

Results and procedure shall follow a coherent order and under appropriate subheadings.

That is a correct remark. We reordered the Results section and added subheadings as suggested by the reviewer. This indeed improved this section's coherency. The paragraphs in the Results section are now organized in the following order:

  • An opening paragraph specifies the research aims and experimental logic (Line 278).
  • 1. Trichoderma spp. late wilt control in the 2019 growing season (line 286).

       This section was edited and improved.

  • 2. Trichoderma spp. late wilt control in the 2020 growing season (line 318).
    • 2.1. Enhancing the disease using wooden toothpicks inoculation

Newly added subheading.

  • 2.2. The 2020 experiment results at the sprouting growth phase

Newly added subheading

  • 2.3. The 2020 experiment results at the harvest

Newly added subheading.

  • 2.4. Lower stem symptoms evaluation in the 2020 growing season

Newly added subheading.

  • 2.5. Cobs’ spathes symptoms evaluation in the 2020 growing season

Newly added subheading.

  • 2.6. M. maydis DNA evaluation at the 2020 season’s end

Newly added subheading.

Discussion is very long and loses the sense of research in middle. Shall be rewritten in a coherent order substantiating with references not old as 2010.

We edited and improved the Discussion according to the reviewer's suggestions and remarks.

Regarding the replacement of old references, we corrected this. Still, five references could not be substituted with new citations since no further studies have been conducted on the subject. Late wilt disease has been reported so far in only eight countries and is considered an exotic and unfamiliar disease in most parts of the world. Consequently, only a few scientific studies are published on the subject annually.

The following changes were made to the Discussion:

  • The following paragraph was removed from the text: “We have deliberately infected the experimental field in these two years of research. Still, the field does not have a long history of LWD that characterizes some commercial fields in the area (Hula Valley, north Israel). Apparently, this is why in both seasons (2019 and 2020), the pathogen DNA levels (measured at the harvest in the first aboveground internode) were low in the treatments. These levels in the infected plots (without the stabbing inoculation) were 5.4∗10-4 and 1.3∗10-3 in the 2019’s and 2020’s experiments. At low levels of DNA (usually lower than 1∗10-3), the pathogen’s DNA variations are not always reflected in the plants’ growth parameters and the disease’s symptom severity. Indeed, the two seasons were varied in the disease severity (that was more evident in 2019) and environmental conditions (that were less favorable for the disease development in 2020). These seasonal variations were not reflected in the maydis overall DNA levels.”

  • The following paragraph was edited and rewritten (lines 484-490): “Environmental conditions play a pivotal role in the burst and harshness of plant diseases in the familiar phytopathology triangle model, including also the pathogen and the host. The 2020 experiment was conducted in autumn for practical reasons (mainly the availability of the field). This season was, as expected, colder and rainier, and these conditions were probably the cause of the low yields and a minor outbreak of LWD. Indeed, early sowing of corn in Egypt reduced LWD [45], while late summer planting reduced disease severity in India [46].”

  • The following paragraph was edited and rewritten (lines 491-501): “While that situation may not be ideal for studying preventive treatments, it provides us with a unique opportunity to explore the pathogenesis under conditions that do not allow the appearance of prominent symptoms. Indeed, the plants seemed healthy and vivid in all plots. Nonetheless, close inspection of the lower stem and the cobs’ spathes revealed a different picture, with up to 80% and 60% symptomatic plants, respectively. Tracking the pathogen DNA inside the plants’ roots support these data, with proximity between the DNA levels and symptoms evaluation. Following maydis DNA could provide additional important information. For example, the stalk-stabbing inoculation bypassed the protection afforded by the Trichoderma treatments in the soil, resulting in a high increase in fungal DNA in those treatments. The symptoms’ evaluation supported this conclusion.”

  • The following paragraph was removed from the text: “The rain that accompanies the fall season of 2020 can also be an influential factor. Early sowing of corn in Egypt reduced LWD [46], while late summer planting reduced disease severity in India [47]. Indeed low water potential is considered one of the most influential factors enhancing LWD progression [48,49]. Our own experience [50] and scientific data from other researchers suggests that the first appearance and progression of LWD symptoms are subject to host physiology and environmental changes. In susceptible maize cultivars, the first symptoms of wilting usually appeared in the field 50-60 days after sowing, shortly before the tasseling stage [30,51]. Nevertheless, when drip irrigation was used in a line for each row format (which provides better irrigation) instead of a frontal irrigation system, plants reached 70% silk 60 days after sowing, and the first disease symptoms appeared only 10 days later (70 days after seeding) [52].”
  • The following paragraph was edited and rewritten (lines 534-547): “Low water potential is considered one of the most influential factors enhancing LWD progression [48,49]. maydis is sensitive to low oxygen conditions in wet soils [49]. In contrast, a high oxygen atmosphere promotes the pathogen's colonies’ growth [50]. Reviewing the literature concluded that frequent watering or saturated soils reduced late wilt by influencing the plant’s surrounding resistance to the pathogen and the plant’s immunity. The soil microorganisms’ communities that antagonize M. maydis may be influenced by excessive moisture conditions. Indeed, water availability may be the most central environmental factor affecting the soil’s microbial community and activities [51]. Floods may increase anaerobic conditions that stimulate lytic microorganisms to degrade the pathogen’s sclerotia and reduce its survival potential. There are several supporting pieces of evidence for this line of thinking. M. maydis is considered a poor competitive saprophyte compared to other microorganisms in the soil [7]. Moreover, corn did not develop late wilt following paddy-cultivated rice, which increases the availability of Mn for subsequent crops.”

The paragraphs in the Discussion section are now organized in the following order:

  • Global maize production importance and the risk of diseases including LWD
  • The current study’s aims and logic
  • The highlight (most significant result) of this study
  • The relations between the pathogen’s DNA, plant growth parameters, and disease’s symptom severity
  • The environmental condition’s pivotal role in the severity of disease outbreaks
  • Comparison of the meteorological data and maize LWD in northern Israel
  • Frequent watering or saturated soils reduced late wilt by influencing the plant’s surrounding resistance to the pathogen and the plant’s immunity
  • The application method of asperelloides (T203) could lead to its ineffectiveness in the current study
  • Future green application based on longibrachiatum (T7407) and T. asperellum (P1) as a bioprotective shield against M. maydis may be beneficial if the two species would be applied together
  • The benefit of using asperellum (P1, an endophyte) as a biocontrol agent

In spite of a very interesting and promising research work, the presentation of the work in the manuscript, right from abstract to conclusion is dragging; the discussion section makes it hard to perceive a better understanding of the findings. Discussion section shall be attempted again with a clear flow of correlating the need of experiment, result and outcome – correlating with previous similar citations. The extensive work done by the researchers are not carried out properly in the manuscript. The whole manuscript shall be rewritten in a precise and more understanding manner.

We made our best effort to address this issue. We believe that the new, improved manuscript version reflects this. The major revisions made throughout the manuscript and described in detail above. Additionally, the following changes to the text were made:

  • The following paragraph was removed from the Introduction: “ maydis can undergo pathogenic variations [20]. Thus, when applying biocontrol antagonists, they change along with the pathogen variability in a continuous evolutionary arms race.”
  • The following paragraph was removed from the Introduction: “This molecular-based technique allows detecting the pathogen DNA in plant tissues at a concentration difference of million times [18].”
  • The following paragraph was added to the Conclusions (lines 619-622): “Such a solution would require pre-testing the effect of these chemical fungicides on the Trichoderma species since fungicides could reduce their impact. If adequately developed, such an integrative control method’s main benefit would be reducing the use of pesticides.”

The manuscript shall also be checked for typographical and formatting errors.

A professional English scientific copy editor edited the entire manuscript.

Round 2

Reviewer 1 Report

Thank you for improving the manuscript according to the highlighted questions.

Author Response

We want to express our sincere appreciation to the reviewer for essential and helpful advice. The time and effort invested are greatly appreciated and certainly contributed to the manuscript and improved it. Thank you.

Reviewer 2 Report

All the corrections mentioned earlier are carried out satisfactorily. Yet there are many typographical and formatting errors, like not italicizing scientific names. It is minor.

 Please reformat

 Page 4 - lines 138-40, they are complex and unable to understand - kindly change to a simpler form

Pg 4 - line 146 - it is or control or and control.

Pg 4 - line 162 - change citation format.

Pg 5 line 171-180 - rewrite the paragraph - what does see reports from Spain [13,34], Egypt [14] and Israel [6,30] mean???

Consider changing subsection 2.5.2. heading.

Please remove table/figure references in discussion.

Please change section 5 - conclusions to conclusion and reduce conclusion to one paragraph. Please ensure references are in correct and uniform format.

The reference section should be updated and discuss more about biopesticide through microbial culture with recent updates in the introduction as well in the discussion at least the author should refer the following documents and rewrite the advantages of biopesticides. refer 10.1080/03235408.2018.1496525, 10.1016/j.jip.2018.10.008, 10.1016/j.ecoenv.2019.109474, 0.1007/978-81-322-2056-5_3.

I would redirect the manuscript for publication after carry out the minor revision

Author Response

Responses to Reviewer 2’s comments

We thank the reviewer for investing substantial efforts, which are undoubtedly contributing to this manuscript. The remarks and suggestions improved this paper’s scientific soundness and accurateness.

All the corrections mentioned earlier are carried out satisfactorily. Yet, there are many typographical and formatting errors, like not italicizing scientific names. It is minor. Please reformat

The reviewer is correct; this issue should be double-checked. We made sure that all the formatting errors, like scientific names, are accurate and correct.

Page 4 - lines 138-40, they are complex and unable to understand - kindly change to a simpler form

We agree, the sentence was rewritten as suggested by the reviewer, and it now reads: “The experiment field has a record of moderate LWD infection. We deliberately inoculated the experiment plots (except the control plots) to achieve a more severe disease burst.

Pg 4 - line 146 - it is or control or and control.

Corrected to “and control”.

Pg 4 - line 162 - change citation format.

The sentence was corrected, and it now reads: “The plant growth methodology and inoculation were similar to [10,17].”

Pg 5 line 171-180 - rewrite the paragraph - what does see reports from Spain [13,34], Egypt [14], and Israel [6,30] mean???

The paragraph was rewritten as advised by the reviewer: “A naturally infested field is populated with pathogen lines having different aggressiveness, as reported in Spain [13,34], Egypt [14], and Israel [6,30].”

Consider changing subsection 2.5.2. heading.

Subsection 2.5.2. heading corrected to: “The 2020 experiment”.

Please remove table/figure references in the Discussion.

There are no references to Figures in the Discussion. There is one reference to Table 7, which is presented in the Discussion.

Please change section 5 - conclusions to conclusion and reduce conclusion to one paragraph. Please ensure references are in a correct and uniform format.

Section 5 headline was corrected to “conclusion” as advised.

The conclusion section was shortened to one paragraph as advised.

The reference section should be updated and discuss more about biopesticide through microbial culture with recent updates in the introduction as well in the Discussion at least the author should refer to the following documents and rewrite the advantages of biopesticides. refer 10.1080/03235408.2018.1496525, 10.1016/j.jip.2018.10.008, 10.1016/j.ecoenv.2019.109474, 0.1007/978-81-322-2056-5_3.

We agree with the reviewer. The following new paragraph regarding biopesticide control was added to the Discussion section, based on the suggested references (lines 597-606): “

Integrated LWD control strategy can combine the Trichoderma species inspected here with other environmentally friendly solutions such as biopesticide that are gaining increasing importance [54]. In India, for instance, microbial biopesticides research is evolving rapidly, and such pesticides comprise ≈ 5% of the market [55]. Biopesticides developed from pathogenic viruses, bacteria, fungi, nematodes, and plants’ secondary metabolites, are alternatives to chemical pesticides and are a significant component of many pest control programs [56]. New bacteria-based biopesticides are constantly developed. For example, it was shown that volatile compounds and peptides from the bacteria Bacillus subtilisStaphylococcus aureus, and Pseudomonas aeruginosa inhibited the hyphal growth and melanin production of A. solani [57].”
